# Assessing geographical variation in ovulatory cycle knowledge among women of reproductive age in Sierra Leone: Analysis of the 2019 Demographic and Health Survey

Edward Kwabena Ameyaw[1], Daniel Woytowich[2], Fred Yao Gbagbo[3‡]*, Padmore Adusei Amoah[1,4‡]

1 Institute of Policy Studies and School of Graduate Studies, Lingnan University, Tuen Mun, Hong Kong, 2 California State University Los Angeles, Los Angeles, California, United States of America, 3 University of Education, Winneba, Ghana, 4 Department of Psychology, School of Graduate Studies, Institute of Policy Studies, Lingnan University, Tuen Mun, Hong Kong SAR

☯ These authors contributed equally to this work.
‡ FYG and PAA also contributed equally to this work.
* gbagbofredyao2002@yahoo.co.uk, fygbagbo@uew.edu.gh

**Data Availability Statement:** All data is available from the Statistical Services of Sierra Leone.

## Abstract

### Background

Sierra Leone has poor indicators of reproductive health and a high prevalence of unintended pregnancies. To date, no study has explored determinants of ovulatory cycle knowledge in Sierra Leone. We investigated geographic region to determine where the needs for improved ovulatory cycle knowledge are greatest in Sierra Leone.

### Methods

This is a cross-sectional study of women of reproductive age (n = 15,574) based on the 2019 Sierra Leone Demographic and Health Survey. Geographic region and sociodemographic covariates were included in a multivariate logistic regression model predicting the odds that participants possessed accurate knowledge of when in the ovulatory cycle pregnancy initiation is most likely.

### Results

In Sierra Leone, 39.8% (CI = 37.4–40.9) of 15-49-year-old women had accurate knowledge of the ovulatory cycle. Women in the Northern and Southern regions possessed the highest prevalence of correct knowledge (46.7%, CI = 43.1–50.3 and 45.1%, CI = 41.9–48.2, respectively). Women from the Northwestern (AOR = 0.29, CI = 0.22–0.38), Eastern (AOR = 0.55, CI = 0.41–0.72), and Western regions (AOR = 0.63, CI = 0.50–0.80) had significantly lower odds of accurate ovulatory cycle knowledge compared to others. Women aged 15–19, those with a primary school education, and participants with a parity of none all had the lowest odds of correct ovulatory cycle knowledge as well.

**Funding:** The authors received no specific funding for this work.

**Competing interests:** The authors have declared that no competing interests exist.

## Conclusion

Less than four in ten women in Sierra Leone had accurate knowledge of when in the ovulatory cycle pregnancy is most likely to occur. This suggests that family planning outreach programs should include education on the ovulatory cycle and the importance of understanding the implications of its timing. This can reduce the risk of unintended pregnancies throughout Sierra Leone, and can have an especially positive impact in the Northwestern, Eastern, and Western regions, where ovulatory cycle knowledge was significantly lower.

## Background

During ovulation, the major follicle ruptures and releases an egg into the fallopian tube where fertilization can occur if sperm is present [1]. An understanding of this relationship between ovulation and pregnancy initiation, and accurate knowledge of when in the menstrual cycle ovulation occurs, are both invaluable pieces of family planning information [2, 3]. Ovulation generally occurs at about the midpoint of the menstrual cycle. Therefore, for the average menstrual cycle of 28 days, ovulation will occur about 14 days after the onset of menstruation (i.e., when the period begins) [4]. It is important to note however that there can be variation in menstrual cycle length, which means that the occurrence of ovulation can vary from the typical 14-day mark [5]. In any case, menstruation can be used to estimate when one will ovulate [3]. Women who have a basic knowledge of the aforementioned physiological phenomena are better able to time their pregnancies, or avoid pregnancy if that is their goal [2, 3].

Indicators of ovulation can include mood changes, increases in basal body temperature, the increased production of clear and non-viscous cervical mucous, among others [3, 6, 7]. Women wishing to become pregnant can increase their chances of doing so by having unprotected intercourse in the days leading up to ovulation, while women not wishing to become pregnant should at a minimum have protected sex, ensure the proper use of another birth control method, or abstain from sex altogether during this timeframe [8–10]. When ovulation occurs in relation to when unprotected sex was had will also dictate how effective levonorgestrel-based emergency contraception will be [11]. Inadequate knowledge of the ovulatory cycle is therefore a prime determinant of unintended and mistimed pregnancies [12–15], which carry with them heightened risks of miscarriages, abortions, stillbirths, and childhood morbidity [16–20].

Despite the benefits of being familiar with one's ovulatory cycle, empirical evidence shows that accurate knowledge of the ovulatory cycle is generally low worldwide [12, 21–24]. Part of the reason for this may lie in the fact that discussions regarding menstrual cycle issues are still taboo for many individuals, families, and societies [21, 25]. In Sub-Saharan Africa, investigations into the proportion of women with correct ovulatory cycle knowledge have produced varied results, but again show relatively low prevalence of accurate knowledge [15, 26]. The prevalence of correct knowledge was as low as 10.4% in São Tomé and Príncipe and as high as 49.0% in Comoros in one multi-country assessment of 15–24-year-olds [15]. The same study revealed that in 2013, 15–24-year-old women in Sierra Leone had a 30.3% prevalence of proper ovulatory cycle knowledge [15].

In addition to low ovulatory cycle knowledge, Sierra Leone has some of the worst maternal and pregnancy-related health indicators in the world. The maternal mortality rate in Sierra Leone (1,360 per 100,000 live births) is the highest in the world [27]. Their neonatal mortality rate (34 per 1,000 live births) is among the highest in the world as well [27]. Also notable is the

fact that abortion is illegal in Sierra Leone unless it is needed to save the mother's life [28]. All of these issues make avoiding unintended pregnancies in Sierra Leone especially important, of which having correct ovulatory cycle knowledge is an integral part. Unfortunately, to our knowledge, there are no studies about the association between geographic and other sociodemographic factors with ovulatory cycle knowledge in Sierra Leone. Research on this subject is a necessity if a complement of factors that may predispose women to a lower prevalence of ovulatory cycle knowledge, which places them and their offspring at risk, are to be elucidated [29–31].

To begin to fill this knowledge gap, this study will assess the association between Sierra Leone's five regions (Eastern province, Northern province, Northwestern province, Southern province, and Western area (Fig 1) and other covariates with correct ovulatory cycle knowledge. Understanding which regions may have a higher or lower prevalence of accurate knowledge, and also understanding if region alone is maintained as a significant predictor in a multivariate model can help in more efficiently targeting family planning educational resources in Sierra Leone on a geographical basis. It can also provide insight into the regions of Sierra Leone that may be at higher risk of unintended pregnancies, as well as catalyze subsequent research into sociocultural covariates that may lead to differing levels of ovulatory cycle

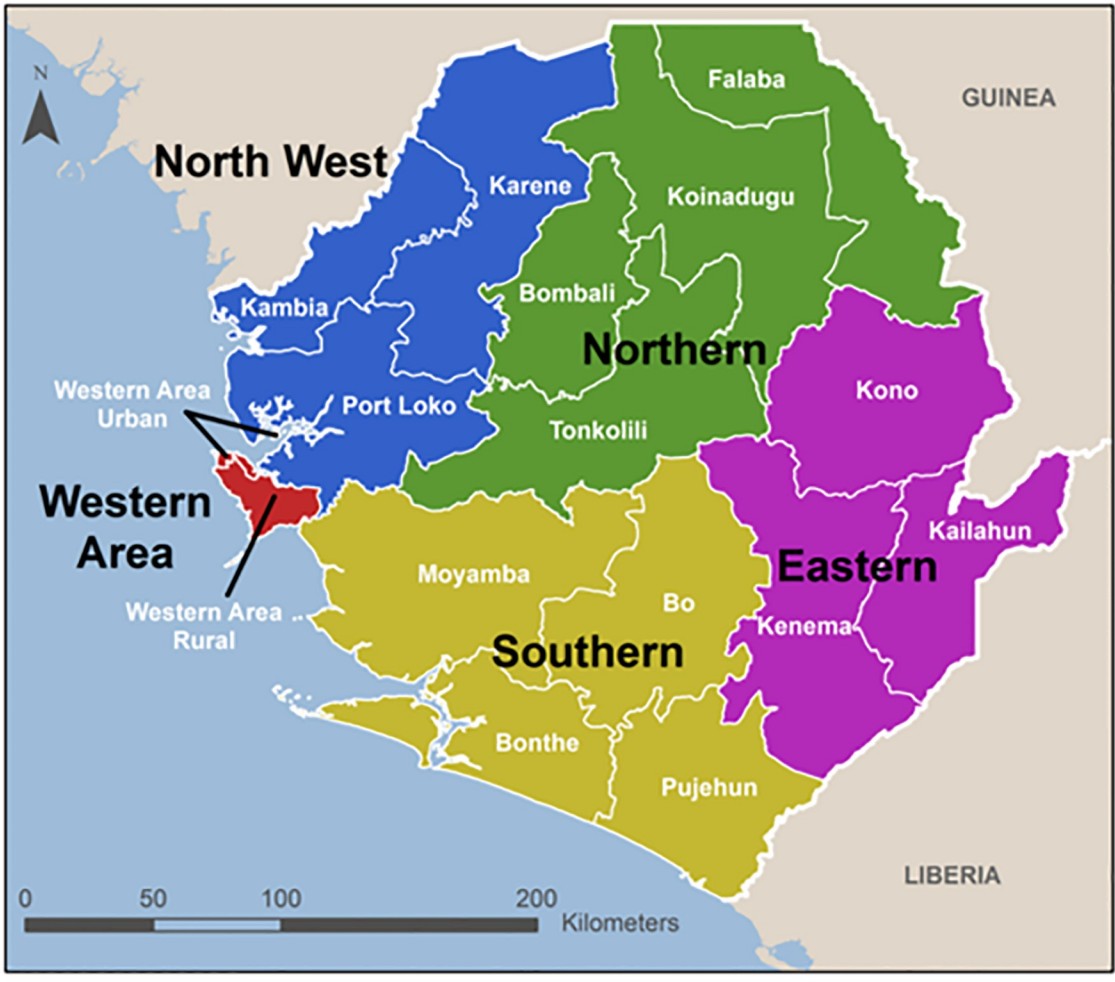

**Fig 1. The map showing regions of Sierra Leone here.**

knowledge. All these have policy and programme implications to chart a path forward in the area of enhanced sexual and reproductive health of women in reproductive ages in Sierra Leone.

## Methods

### Study design

This is a quantitative cross-sectional study that uses secondary data from the 2019 Demographic and Health Survey (DHS) of Sierra Leone [32].

### Data source

Permission to use the dataset was obtained from the Measure DHS Program [33]. The 2019 Sierra Leone survey was part of DHS Version VII (i.e., DHS surveys are conducted roughly every 5 years, so version VII is the seventh iteration). The final report of the 2019 Sierra Leone DHS can be found here: https://dhsprogram.com/publications/publication-FR365-DHS-Final-Reports.cfm; while the full dataset can be downloaded here: https://dhsprogram.com/data/available-datasets.cfm. DHS are nationally representative household surveys that utilize a two-stage stratified cluster sampling design. DHS surveys employ standardized data collection procedures which allow for consistency and comparability across regions and populations, generally have response rates of over 90%, accurately represent marginalized groups with complex sampling techniques, and provide up-to-date and comprehensive training for interviewers [34]. Survey techniques and protocols were approved by both ICF International, who provided technical assistance through The DHS Program, and Statistics Sierra Leone, who implemented the survey in-country on behalf of the Sierra Leone Ministry of Health and Sanitation. Funding for the 2019 Sierra Leone DHS was provided by the United States Agency for International Development, the Global Fund, Department for International Development, the United Nations Population Fund, World Health Organization, and World Bank.

### Sampling

In the first sampling stage, enumeration areas were drawn from the national census according to population and stratification characteristics. In the second stage, systematic sampling was used to select households from enumeration areas where the interviews would take place [35]. In all subsequent calculations, sampling weights were applied to adjust for DHS's complex survey design. Further information about sampling, weighting, and other DHS methodologies can be found in the Guide to DHS Statistics, DHS-7, Version 2 [35]. Out of the 13,399 households enlisted for the survey, data was collected from 15,574 respondents [32]. The survey interviews were conducted from May to August 2019 [32].

### Inclusion criteria

The respondents were women of reproductive age (15–49 years) at the time of the survey who had provided information on all variables required for the analysis.

### Outcome variable

The outcome of interest was captured by the DHS variable 'V217', which asked respondents which point in the ovulatory cycle presented the highest risk of pregnancy. Possible answers to the originally worded question were: 'during her period', 'after period ended', 'middle of the cycle', 'before the period begins', 'at any time', 'other', and 'don't know'. We recoded this into

a binary variable reflective of correct knowledge of the ovulatory cycle by recoding responses of 'middle of the cycle' into '1', meaning those respondents had correct knowledge of the ovulatory cycle; and all other responses into '0', meaning those respondents did not have accurate knowledge of the ovulatory cycle. This method of recoding the 'V217' DHS variable has been used in other notable studies assessing the prevalence of accurate ovulatory cycle knowledge [29–31].

### Independent variables

The main predicting variable was regional category, consisting of 'Eastern', 'Northern', 'North-western', 'Southern', and 'Western' regions of Sierra Leone. These are illustrated in Fig 1, which depicts the map showing the Regions of Sierra Leone.

In addition to the Sierra Leonean region, twelve other sociodemographic covariates were selected according to previous literature and epidemiological plausibility [29–31]. These included age (15–19, 20–24, 25–29, 30–34, 35–39, 40–44, and 45–49 years), type of residence (urban, rural), religion (Christian, Islam), wealth index (poorest, poorer, middle, richer, richest), highest education level achieved (no education, primary, secondary, higher), marital status (never married or in the union, married or cohabiting, widowed/divorced/separated), parity (none, 1–2, 3–4, >5 or more), contraception knowledge (knows no method, knows folkloric method, knows traditional method, knows modern method), contraceptive use and intention (currently using modern contraception, currently using a traditional method, not using contraception but intends to use at a later date, never intends to use contraception), and the frequency of reading newspapers/magazines, listening to the radio, and watching television (not at all, less than once a week, at least once a week).

### Data analysis

The percentage of respondents with accurate knowledge of the ovulatory cycle was estimated across the regions of Sierra Leone and all aforementioned covariates in a descriptive analysis. The statistical significance of the bivariate association between the independent variables and the outcome variable was initially assessed using chi-square ($\chi2$) tests of independence. Next, a bivariate logistic regression model (Model 1) was used to produce an unadjusted odd ratio (OR) of the association between living in one of the five Sierra Leone regions and correct ovulatory cycle knowledge. Model 2 was a multivariate logistic regression model in which all sociodemographic variables that had statistically significant ($p<0.05$) bivariate associations in initial tests of independence were included and controlled for. Model 2 produced adjusted odds ratios (aORs) and corresponding 95% confidence intervals (CI) that reflected their precision and significance. Incorrect ovulatory cycle knowledge was the reference group meaning that an OR greater than one (1) indicated that respondents in that corresponding category had a higher likelihood of possessing correct ovulatory cycle knowledge. We performed sub-group analyses by conducting a multivariable logistic regression on samples from each of the administrative regions as well. Missing data were dropped because it was minimal (less than 2%). All statistical analyses were carried out using Stata Version 17 and sample weights were applied using procedures outlined in the Guide to DHS Statistics, DHS-7, Version 2 [35].

### Ethics approval

Our study reports on research involving human participants, however, ethical approval was not applicable for our secondary analysis since The DHS Program already had ethical clearance for conducting the primary survey and made the data available for use by the public. Because we strictly followed the conditions of the institutional review board that approved the

DHS in Sierra Leone, further ethical approval was not required. Additionally, our methodology did not require informed consent from participants because informed consent was already provided to the original implementers of the survey. We did request permission from The DHS Program to use the 2019 Sierra Leone DHS dataset, which was approved before we began our analysis.

## Results

### Descriptive results

Table 1 shows the raw counts and weighted percentages of respondents across all the sociodemographic variables of interest. Out of the total sample (n = 15,574), 39.8% (CI = 37.4–40.9) had correct knowledge of the ovulatory cycle. Regarding region, women from the Northern region of Sierra Leone had the highest prevalence of correct ovulatory cycle knowledge (46.7%, CI = 43.1–50.3), whereas women from the Northwestern region had the lowest (20.1%, CI = 16.7–24.0), followed by women from the Eastern (33.3%, CI = 28.5–38.5), Western (39.2%, CI = 35.4–43.2), and Southern regions (45.1%, CI = 41.9–48.2). Women aged 15–19 had the lowest percentage (27.8%, CI = 25.5–30.2) of ovulatory cycle knowledge whereas 35–39-year-olds had the highest (42.1%, CI = 39.3–44.9). Urban women had slightly higher (39.1%, CI = 36.2–42.0) ovulatory cycle knowledge than rural women (36.5%, CI = 34.3–38.6); however, there was no statistically significant difference between the two (p = 0.153). Religion (p = 0.102) and wealth index (p = 0.530) also had no significant bivariate associations with ovulatory cycle knowledge across categories.

Women with no education (36.3%, CI = 34.3–38.3) and primary educations (31.2%, CI = 28.6–33.9) had the lowest prevalence of ovulatory cycle knowledge whereas those with 'higher' educations had the highest (54.1%, CI = 48.0–60.0). Respondents who were never married/in a union had the lowest percentage of correct ovulatory cycle knowledge (34.3%, CI = 32.0–36.7) while those married/cohabitating had the highest (39.4%, CI = 37.5–41.3). Respondents with a parity of none had the lowest percentage of correct ovulatory cycle knowledge (30.9%, CI = 28.6–33.3) whereas respondents with 1–2 prior births (41.0%, CI = 38.7–43.4) and 3–4 prior births (41.0%, CI = 38.6–43.4) had the highest.

Contraception knowledge had no association with ovulatory cycle knowledge (p = 0.396). As for current contraceptive use and intention, respondents who reported never intending to use contraception had the lowest prevalence of correct ovulatory cycle knowledge (33.1%, CI = 30.8–35.4), whereas women currently using a traditional method of contraception had the highest (49.6%, CI = 31.7–67.6). Lastly, women who read the newspaper/magazine at least once a week (51.5%, CI = 44.5–58.4) and/or listened to the radio at least once a week (43.2%, CI = 39.9–46.5) had a higher prevalence of ovulatory cycle knowledge than women who engaged in those respective activities less often. Frequency of watching television did not have an association with ovulatory cycle knowledge (p = 0.457).

### Logistic regression results

Table 2 shows the logistic regression results for the bivariate unadjusted model (Model 1) and the multivariate adjusted model (Model 2). In Model 1, only the main predictor of interest (region) was assessed for its association with ovulatory cycle knowledge. Women of the Northwestern region, compared to the reference group of Northern region women, had the lowest odds of correct ovulatory cycle knowledge (OR = 0.29, CI = 0.22–0.38), followed by women of the Eastern region (OR = 0.57, CI = 0.43–0.74) and Western region (OR = 0.74, CI = 0.59–0.92), while women of the Southern region did not have significantly different odds of ovulatory cycle knowledge (OR = 0.94, CI = 0.77–1.13) compared to the reference group. We also

**Table 1.  Accurate ovulatory cycle knowledge stratified by sociodemographic characteristics.**

| Sociodemographic Characteristics | Frequency (%) | Percentage with correct knowledge of ovulatory cycle (95% CI) | $X^2$ (p-value) |
|---|---|---|---|
| **Whole Sample (2019 Sierra Leone DHS)** | 15574 | 39.8 (37.4–40.9) | - |
| **Region of Sierra Leone** | | | $X^2$ = 23.9 (<0.001) |
| Eastern | 2978 (19.7) | 33.3 (28.5–38.5) | |
| Northern | 3971 (21.3) | 46.7 (43.1–50.3) | |
| Northwestern | 2498 (16.1) | 20.1 (16.7–24.0) | |
| Southern | 3513 (18.6) | 45.1 (41.9–48.2) | |
| Western | 2614 (24.3) | 39.2 (35.4–43.2) | |
| **Age (years)** | | | $X^2$ = 22.1 (<0.001) |
| 15–19 | 3460 (22.0) | 27.8 (25.5–30.2) | |
| 20–24 | 2602 (16.9) | 40.2 (37.6–42.9) | |
| 25–29 | 2619 (17.5) | 41.6 (38.8–44.5) | |
| 30–34 | 1963 (12.5) | 40.4 (37.5–43.4) | |
| 35–39 | 2251 (14.3) | 42.1 (39.3–44.9) | |
| 40–44 | 1358 (8.6) | 37.2 (33.7–40.7) | |
| 45–49 | 1321 (8.3) | 39.0 (35.3–42.8) | |
| **Type of Residence** | | | $X^2$ = 2.05 (0.153) |
| Urban | 6399 (46.0) | 39.1 (36.2–42.0) | |
| Rural | 9175 (54.0) | 36.5 (34.3–38.6) | |
| **Religion** | | | $X^2$ = 2.67 (0.102) |
| Christian | 3546 (23.2) | 38.4 (35.5–41.4) | |
| Islam | 12021 (76.8) | 37.4 (35.6–39.3) | |
| **Wealth Index** | | | $X^2$ = 0.40 (0.530) |
| Poorest | 3077 (17.6) | 36.1 (33.1–39.2) | |
| Poorer | 3022 (18.2) | 36.4 (33.4–39.4) | |
| Middle | 3190 (19.0) | 37.0 (34.1–40.1) | |
| Richer | 3366 (21.7) | 40.6 (37.5–43.8) | |
| Richest | 2919 (23.5) | 37.6 (34.0–41.3) | |
| **Highest Education Level Achieved** | | | $X^2$ = 21.92 (<0.001) |
| No education | 7535 (45.5) | 36.3 (34.3–38.3) | |
| Primary School | 2034 (13.5) | 31.2 (28.6–33.9) | |
| Secondary School | 5419 (36.8) | 39.8 (37.3–42.3) | |
| Higher | 586 (4.3) | 54.1 (48.0–60.0) | |
| **Marital Status** | | | $X^2$ = 10.55 (<0.001) |
| Never married or in union | 4966 (32.5) | 34.3 (32.0–36.7) | |
| Married and/or living with partner | 9837 (62.4) | 39.4 (37.5–41.3) | |
| Widowed/Divorced/Separated | 771 (5.1) | 37.9 (33.6–42.4) | |
| **Parity** | | | $X^2$ = 26.40 (<0.001) |
| None | 4099 (26.4) | 30.9 (28.6–33.3) | |
| 1–2 | 4523 (30.1) | 41.0 (38.7–43.4) | |
| 3–4 | 3543 (22.3) | 41.0 (38.6–43.4) | |
| 5 or more | 3409 (21.1) | 37.8 (35.3–40.4) | |
| **Contraception Knowledge** | | | $X^2$ = 0.978 (0.396) |
| Knows no method of contraception | 519 (2.4) | 40.5 (33.9–47.5) | |
| Knows only folkloric method | 16 (0.1) | 21.8 (5.6–56.9) | |
| Knows only traditional method | 17 (0.1) | 23.7 (9.9–46.9) | |
| Knows modern method | 15022 (97.4) | 37.6 (35.8–39.4) | |
| **Contraceptive Use and Intention** | | | $X^2$ = 17.53 (<0.001) |

*(Continued)*

**Table 1.** (Continued)

| Sociodemographic Characteristics | Frequency (%) | Percentage with correct knowledge of ovulatory cycle (95% CI) | $X^2$ (p-value) |
|---|---|---|---|
| Currently using modern method | 3662 (23.9) | 43.2 (40.7–45.7) | |
| Currently using traditional method | 51 (0.3) | 49.6 (31.7–67.6) | |
| Not using contraception but intends to use at a later date | 5783 (37.9) | 38.6 (36.4–40.9) | |
| Never intends to use contraception | 6078 (37.8) | 33.1 (30.8–35.4) | |
| **Frequency of reading newspaper/magazine** | | | $X^2 = 10.92$ (<0.001) |
| Not at all | 14503 (92.0) | 36.9 (35.1–38.7) | |
| Less than once a week | 717 (5.5) | 44.2 (38.5–50.1) | |
| At least once a week | 346 (2.5) | 51.5 (44.5–58.4) | |
| **Frequency of listening to radio** | | | $X^2 = 10.21$ (<0.001) |
| Not at all | 9119 (55.6) | 35.7 (33.6–37.8) | |
| Less than once a week | 2991 (20.4) | 36.6 (33.6–39.7) | |
| At least once a week | 3464 (24.0) | 43.2 (39.9–46.5) | |
| **Frequency of watching television** | | | $X^2 = 0.77$ (0.457) |
| Not at all | 11717 (71.5) | 37.7 (35.7–39.7) | |
| Less than once a week | 1905 (13.5) | 35.7 (32.1–39.4) | |
| At least once a week | 1952 (14.9) | 39.2 (34.9–43.8) | |

**Source:** Sierra Leone Demographic and Health Survey, 2019

**Notes:** The percentages in the frequency column are column percentages (i.e., percentage of the total sample). Percentages in the 'Percentage with correct knowledge of ovulatory cycle' column are row percentages. "n" values are unweighted counts. Percentages and 95% confidence intervals are weighted. Tests of independence were conducted within columns. The χ2 value is a second-order Rao-Scott adjusted chi-square based on the adjusted F.

performed subgroup analyses for each of the five regions with multivariable regressions. However, no significant variation was observed relative to our model results, which is being interpreted in this section.

In bivariate tests of independence (Table 1), age, education level, marital status, parity, contraceptive use and intention, frequency of reading the newspaper/magazine, and frequency of listening to the radio, in addition to region, all had statistically significant associations (p<0.001) with ovulatory cycle knowledge. Therefore, these variables were included in the Model 2 regression model. In Model 2, women living in the Northwestern region had the lowest prevalence of ovulatory cycle knowledge (aOR = 0.29, CI = 0.22–0.38), followed by those in the Eastern (aOR = 0.55, CI = 0.41–0.72) and Western regions (aOR = 0.63, CI = 0.50–0.80) as compared to women from the Northern region. Regarding age, women from older age groups (40–44 and 45–49 years) did not have statistically different aORs compared to the reference group of 35–39-year-olds. Women from younger age groups had significantly lower odds of correct ovulatory cycle knowledge, with those from the 15–19-year age category demonstrating the lowest (aOR = 0.51, CI = 0.41–0.63).

Women with no education (aOR = 0.48, CI = 0.38–0.62), primary education (aOR = 0.44, CI = 0.33–0.57), and secondary education (aOR = 0.72, CI = 0.57–0.92) had significantly lower odds of ovulatory cycle knowledge than those with 'higher' educations (reference group). In Model 2, no marital status category had significantly different odds of having correct ovulatory cycle knowledge. Regarding parity, the only category with significantly lower odds of possessing correct ovulatory cycle knowledge were women with a parity of none (aOR = 0.75, CI = 0.65–0.87), and there was no significant difference in odds between those with 1–2 previous births (reference group), 3–4 previous births (aOR = 1.01, CI = 0.89–1.16), and 5 or more previous births (aOR = 0.97, CI = 0.83–1.13). Categories of contraceptive use

**Table 2. Bivariate and multivariate logistic regression results for determinants of ovulatory cycle knowledge.**

| Sociodemographic Characteristics | Model 1 | Model 2 |
|---|---|---|
| | OR (95% CI) | AOR (95% CI) |
| **Region of Sierra Leone** | | |
| Eastern | 0.57***(0.43–0.74) | 0.55*** (0.41–0.72) |
| Northern | Ref | Ref |
| Northwestern | 0.29***(0.22–0.38) | 0.29*** (0.22–0.38) |
| Southern | 0.94 (0.77–1.13) | 0.92 (0.75–1.13) |
| Western | 0.74**(0.59–0.92) | 0.63***(0.50–0.80) |
| **Age (years)** | | |
| 15–19 | | 0.51*** (0.41–0.63) |
| 20–24 | | 0.81* (0.68–0.96) |
| 25–29 | | 0.81** (0.69–0.95) |
| 30–34 | | 0.85* (0.73–0.99) |
| 35–39 | | Ref |
| 40–44 | | 0.86 (0.73–1.02) |
| 45–49 | | 0.99 (0.82–1.21) |
| **Highest Education Level Achieved** | | |
| No education | | 0.48***(0.38–0.62) |
| Primary School | | 0.44***(0.33–0.57) |
| Secondary School | | 0.72** (0.57–0.92) |
| Higher | | Ref |
| **Marital Status** | | |
| Never married or in union | | 0.98 (0.85–1.14) |
| Married and/or living with partner | | Ref |
| Widowed/Divorced/Separated | | 0.87 (0.71–1.06) |
| **Parity** | | |
| None | | 0.75*** (0.65–0.87) |
| 1–2 | | Ref |
| 3–4 | | 1.01 (0.89–1.16) |
| 5 or more | | 0.97 (0.83–1.13) |
| **Contraceptive Use and Intention** | | |
| Currently using modern method | | 0.82 (0.42–1.61) |
| Currently using traditional method | | Ref |
| Not using contraception but intends to use at a later date | | 0.75 (0.38–1.50) |
| Never intends to use contraception | | 0.57 (0.30–1.12) |
| **Frequency of reading newspaper/ magazine** | | |
| Not at all | | 0.81 (0.60–1.08) |
| Less than once a week | | 0.88 (0.63–1.25) |
| At least once a week | | Ref |
| **Frequency of listening to radio** | | |
| Not at all | | 0.91 (0.79–1.06) |
| Less than once a week | | 0.80*(0.67–0.96) |
| At least once a week | | Ref |

Ref = Reference group,

***p<0.001,

**p<0.01,

*p<0.05

**Notes:** Reference categories were chosen based on which had the highest percentage of correct knowledge from the descriptive analysis.

and intention and frequency of reading the newspaper/magazine became statistically insignificant in Model 2. Regarding the frequency of listening to the radio, women who listened to the radio less than once a week had very slightly lower odds of possessing correct ovulatory cycle knowledge (aOR = 0.80, CI = 0.67–0.96) than those who listened at least once a week (reference group).

## Discussion

This study estimated the prevalence of correct ovulatory cycle knowledge across Sierra Leone and sought to determine if region, along with other sociodemographic factors, was associated with significantly higher or lower levels of that knowledge. In all, 39.8% of 15–49-year-old women in Sierra Leone demonstrated correct knowledge of when in the ovulatory cycle fertilization/pregnancy is most likely. Iyanda and colleagues' [15] findings from nearby countries showed that 23.3%, 11.5%, 23.1%, and 36.6% of 15–24-year-old women from Guinea (2012), Liberia (2013), Gambia (2013), and Burkina Faso (2010), respectively, had accurate ovulatory cycle knowledge. The same study showed that 30.3% of 15–24-year-olds in Sierra Leone in 2013 had correct ovulatory cycle knowledge, compared to our finding of 39.8% for women of all reproductive ages in 2019 [15]. It is possible that more recent percentages from Guinea, Liberia, Gambia, and Burkina Faso from women of all reproductive ages would be higher than Iyanda and colleagues' findings from 15-24-year-olds as well [15]. Getahun and Nigatu [29] found that 23.6% of 15–49-year-old women in Ethiopia in 2016 had accurate ovulatory cycle knowledge.

While intercountry differences are apparent in the aforementioned results, previous studies and ours alike demonstrate a low level of accurate knowledge of the ovulatory cycle among women in Sub-Saharan Africa. This low level of knowledge is a major public health concern since women who lack awareness about when they ovulate, and of the significance of ovulation and its role in fertilization, are more likely to experience unintended pregnancies and their repercussions [12–15]. This is especially important to consider for countries in which abortion is illegal, such as Sierra Leone [28].

Our multivariate analysis showed that the main determinant of interest, region of Sierra Leone, was statistically significant in its association with accurate ovulatory cycle knowledge. Age, educational level, and parity were also statistically significant determinants. Women residing in Sierra Leone's Northwestern, Western, and Eastern regions had significantly lower odds of accurate ovulatory cycle knowledge compared to those in the Northern region after the other sociodemographic variables were factored into the multiple regression. In our initial bivariate test of independence, urban/rural status surprisingly was not associated with ovulatory cycle knowledge and so was not included in the regression model. Previous research has generally shown disparities in health knowledge along the urban/rural divide in Africa [29, 36–39]. Since we found no association between urban/rural status and ovulatory cycle knowledge but did find a significant association between region at the province level and ovulatory cycle knowledge, our results indicate that in Sierra Leone there are other geographically linked influences at play other than the urban/rural divide when accounting for ovulatory cycle awareness. Sociocultural differences and norms that may be associated with regions, such as the influence of religion, local government, gender power disparities, family dynamics, and the resulting ability for certain peoples to have open discussions about issues like family planning may have contributed to the regional differences we observed. Future researchers are encouraged to elucidate other determinants that may shed light on why we observed such large differences in the odds of having correct ovulatory cycle knowledge between Sierra Leone's five geographic regions.

This study also showed that women with no previous births had significantly lower levels of ovulatory cycle knowledge compared to those who had previously given birth (irrespective of the number of births). This result is not surprising and was similar to the aforementioned Ethiopian study where currently pregnant women were found to be more knowledgeable about the ovulatory cycle compared to women who have never conceived [29]. The likely explanation for the association between parity and knowledge of the ovulatory cycle is that women who have already had experience with conception or giving birth are more likely to have interacted with skilled health personnel and/or counselors that would have provided reproductive health information. This finding is another possible reminder of the benefits of health system contact regarding family planning and reproductive health.

Higher education was associated with increased odds of correct ovulatory cycle knowledge among respondents, which is consistent with findings from previous studies [30, 40]. This association can likely be attributed to the fact that education generally has a positive impact on knowledge of health and health-related behavior [41–43]. Education is also a strong social determinant of health since it provides avenues for employment, independence, empowerment, and subsequent increased ability for women to discover and seek out knowledge for themselves [44, 45]. Women of the youngest age group, 15–19-year-olds, had the lowest odds of ovulatory cycle knowledge compared to those of older ages. This pattern of younger age being correlated with lower ovulatory cycle knowledge was generally consistent with findings from similar studies [29, 30].

Finally, the global association between the frequency of radio listening and ovulatory cycle knowledge was weak since women who did not listen to the radio at all did not have significantly lower odds of accurate knowledge. However, women who listened to the radio less than once a week had slightly lower odds of having correct ovulatory cycle knowledge as opposed to those who listened more than once per week. This very weak finding concerning radio listening and the fact that there was no association between ovulatory cycle knowledge and newspaper/magazine reading contradicts the generally accepted paradigm that access to mass media usually increases one's health knowledge [37, 46]. However, this may also simply be an indicator that no information on the ovulatory cycle is currently being provided through these sources in Sierra Leone. Our findings regarding media exposure in Sierra Leone should be explored in more detail.

In summary, less than 4 in 10 women of reproductive age in Sierra Leone possess correct knowledge about their ovulatory cycle and its implications on the chances of becoming pregnant. Region of residence, age, education level, and parity were significantly associated with ovulatory cycle knowledge. Women living in the Northwestern, Eastern, and Western regions, as opposed to Sierra Leone's Northern and Southern provinces, had significantly lower odds of having correct ovulatory cycle knowledge even though Sierra Leone's capital is in the Western region. Women living in Sierra Leone's Northwestern region, as compared to those in the Northern region, had the lowest odds of ovulatory cycle knowledge, followed by women in the Eastern, Western, and Southern regions. The differences in ovulatory cycle knowledge across regions of Sierra Leone suggest that the government and other relevant stakeholders should consider contextually appropriate strategies to enhance the public's knowledge on the ovulatory cycle and its consequences for fertility. Policymakers and community health educators can invest in the development of educational strategies and resources like teach-the-teacher campaigns; easy to understand pamphlets that can be disseminated in communities, schools, and workplaces of healthcare providers; mobile applications that can help women keep track of their cycle; and illustrative videos that outreach workers can use when working with women with low literacy. These resources can be especially targeted to Sierra Leone's Northwestern,

Eastern, and Western areas to minimize the regional disparities in ovulatory cycle knowledge observed in this study.

## Strengths and limitations

A strength of this study is that the DHS dataset used was produced from a complex sampling procedure that ensures appropriate subpopulation representativeness, thereby maximizing the validity of results for our population of interest. The multivariate logistic regression method employed is a reliable way of testing a survey question on knowledge that was easily and accurately dichotomized into 'correct' and 'incorrect' answers. This dataset is also publicly available, which makes our study easily reproducible and/or built upon by other researchers. The same variable on ovulatory cycle knowledge is also available in most other country's DHS surveys, which makes comparable intercountry analyses on this topic possible as well. However, researchers interested in multi-country analyses on the DHS ovulatory cycle knowledge variable will need to consider issues like the year in which the surveys were done, differing reproductive health practices and policies between the countries included, and a myriad of other contextual differences. Also, DHS surveys are replicated roughly every 5 years, which makes a reliable follow-up study of ovulatory cycle knowledge in Sierra Leone possible, which can help researchers and implementers understand how this variable is trending over time. Lastly, this project is the first of its kind, to the best of our knowledge, that has elucidated region of residence and examined other sociodemographic characteristics as predictors of ovulatory cycle knowledge in Sierra Leone. There are several weaknesses to this study as well. One is that it was based on data obtained from a cross-sectional survey that used self-reported measures, which means recall bias must be taken into account. Also, temporal inferences between the sociodemographic covariates and knowledge about the ovulatory cycle could not be ascertained. Social desirability bias often must be considered with sensitive topics like family planning knowledge and practices. The risk of social desirability bias is probably minimal in this study though since there likely would have been no preconceived notion of what the expected answer to the ovulatory cycle question would have been. However, it is possible that respondents that did not know the answer could have guessed in an effort to display that they had some ovulatory cycle knowledge which could have skewed results slightly. Also, while we were able to assess the relationship between residence in Sierra Leone's five broad regions and ovulatory cycle knowledge, which is a necessary first step in looking at geographic influences on ovulatory cycle knowledge, it would have been ideal to also be able to study this topic at an even more granular town or district level. However, that was not possible with this dataset.

## Conclusion

Lack of awareness and understanding about the ovulatory cycle are key contributors to unwanted and mistimed pregnancies. This study found that prevalence of accurate ovulatory cycle knowledge across Sierra Leone is low among women of reproductive age. This is especially concerning since reproductive health indicators are so poor and abortion is illegal in Sierra Leone. In particular, accurate ovulatory cycle knowledge was lowest among 15–19-year-old women, those with primary school educations, those with a parity of none, and in women living in Sierra Leone's Northwestern, Eastern, and Western regions, as opposed to the those living in the Northern and Southern regions. Findings therefore suggest that increased awareness and appreciation of the importance of the ovulatory cycle's role in reproductive health is necessary across all of Sierra Leone, but that resources should especially be targeted towards women belonging to the aforementioned geographic and sociodemographic subgroups.

This epidemiological study augments the current corpus of knowledge and discourse on SSA women's comprehension of the ovulatory cycle and its importance in reproductive health and family planning. Specifically, it contributes to the scientific literature by clearly elucidating a link between geography and ovulatory cycle knowledge among women of reproductive age in Sierra Leone, a phenomenon which has not been documented in the existing literature. This finding provides actionable direction to policymakers and implementers looking to efficiently address geographical imbalances in ovulatory cycle knowledge in Sierra Leone with targeted outreach. Future research can explore the sociocultural mechanisms behind the stark disparities in ovulatory cycle knowledge we observed between Sierra Leone's five regions. Our findings also suggest that studying geographic and sociodemographic determinants of ovulatory cycle knowledge in other countries may provide useful programmatic insights. A more complete understanding of how ovulatory cycle knowledge can be increased in high-risk regions can play an important role in improving reproductive health and decreasing unintended pregnancies throughout the world.

## Supporting information

**S1 Checklist.** *PLOS ONE* **clinical studies checklist.**
(DOCX)

## Acknowledgments

We are grateful to all those who provided information and direction for this paper and to the Measure DHS program.

## Author Contributions

**Conceptualization:** Fred Yao Gbagbo.

**Data curation:** Edward Kwabena Ameyaw, Daniel Woytowich, Padmore Adusei Amoah.

**Formal analysis:** Edward Kwabena Ameyaw, Daniel Woytowich, Padmore Adusei Amoah.

**Methodology:** Fred Yao Gbagbo.

**Supervision:** Padmore Adusei Amoah.

**Writing – original draft:** Fred Yao Gbagbo.

**Writing – review & editing:** Daniel Woytowich, Padmore Adusei Amoah.

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
