## [Decision Letter · Decision Letter 0]

5 Nov 2023

PONE-D-23-11422Assessing the geographical variations in ovulatory cycle knowledge among women of reproductive aged in Sierra Leone: Analysis of the 2019 Demographic and Health SurveyPLOS ONE

Dear Dr. Gbagbo,

Thank you for submitting your manuscript to PLOS ONE. After careful consideration, we feel that it has merit but does not fully meet PLOS ONE’s publication criteria as it currently stands. Therefore, we invite you to submit a revised version of the manuscript that addresses the points raised during the review process.

We look forward to receiving your revised manuscript.

Kind regards,

Obasanjo Afolabi Bolarinwa, Masters

Academic Editor

PLOS ONE

Journal Requirements:

Reviewers' comments:

Reviewer's Responses to Questions

**Comments to the Author**

1. Is the manuscript technically sound, and do the data support the conclusions?

Reviewer #1: Yes

Reviewer #2: Yes

2. Has the statistical analysis been performed appropriately and rigorously? 

Reviewer #1: Yes

Reviewer #2: Yes

3. Have the authors made all data underlying the findings in their manuscript fully available?

Reviewer #1: Yes

Reviewer #2: Yes

4. Is the manuscript presented in an intelligible fashion and written in standard English?

Reviewer #1: Yes

Reviewer #2: Yes

5. Review Comments to the Author

Reviewer #1: This is an interesting project that explores an important area of sexual and reproductive health. The authors have done a great job but a few of these suggestions can make the work better.

Abstract

“We investigated regions of Sierra Leone and other sociodemographic variables to determine where the needs for improved ovulatory cycle knowledge are greatest”.

This appears unclear. “Investigated regions” makes the sentence confusing. Maybe it can be written as: We investigated regions of Sierra Leone where the need for improved ovulatory cycle knowledge is greatest and associated sociodemographic variables.

Conclusion of the abstract: I think “understanding” should be added to the sentence that states: …that stakeholders should integrate comprehensive education on the importance of UNDERSTANDING the ovulatory cycle and its timing, into targeted family planning and reproductive health educational program

Background

Great job on the well-written background section, albeit I have a few comments.

o Indicators of ovulation are not exhaustive, because there might be other signs of ovulation that other women experience that have not been added here, I think the authors should make this obvious by adding “etc”.

o Has there been any similar study on this topic in the context of Sierra Leone? The authors appear very silent on this, if not, aside from informing programs and interventions, I think the authors should state that their study will fill this gap in knowledge.

Method

o Can a weblink for the DHS data be included somewhere in the Data Sources?

o In the data source, the authors should provide information about the data collection agency at the national level in Sierra Leone and any other international or local agencies that provided technical and financial assistance.

o Was there any rationale for not merging the exposure to mass media measured as the frequency of reading newspapers/magazines, listening to the radio, or watching TV? It is common in most studies to see a single measure of exposure to mass media based on these variables.

o Were there missing values in the data? How were they treated in the data analysis?

Results

o In the interpretation of the binary logit regression results, the authors should be careful of anthropomorphism. Hence, I feel the result should read “Women from the Northwestern region compared to the reference group of the Northern region” and other narratives should be directed to the women.

Discussion

o The study limitation/weakness stated by the authors appears too generic, and I do not see any reason why a longitudinal relationship between the region of residence/SES/demographic factors and the outcome interest would be needed.

Reviewer #2: This is an interesting piece that interrogate the geographical variable of level of ovulatory knowledge in Sierra Leone. Please find below a few observation for your attention

1. Arrange the key words alphabetically

2. Introduction

A. Kindly state the rationale for this study

B. What is the gap that necessitates this study

C. Is the issue peculiar to Sierra Leone and SSA?

D. Why was the study conducted in Sierra Leone and not Ghana or Ethiopia?

E. Kindly state the specific objectives

3. Method: show the site where the dataset was curated (under the data source)

What is the meaning of "2019 Sierra Leone survey was part of DHS version II)

4. Results

A. Describe statistics and methods used to examine the subgroups. State the level of interactions between regions

5. Discussion

A. State the policy implications for the study

B. State the contribution to knowledge derived from the results

6. PLOS authors have the option to publish the peer review history of their article (what does this mean?). If published, this will include your full peer review and any attached files.

Reviewer #1: No

Reviewer #2: **Yes: **Chukwudeh Okechukwu Stephen

---

## [Decision Letter · Decision Letter 1]

27 Dec 2023

PONE-D-23-11422R1Assessing the geographical variations in ovulatory cycle knowledge among women of reproductive age in Sierra Leone: Analysis of the 2019 Demographic and Health SurveyPLOS ONE

Dear Dr. Gbagbo,

Thank you for submitting your manuscript to PLOS ONE. After careful consideration, we feel that it has merit but does not fully meet PLOS ONE’s publication criteria as it currently stands. Therefore, we invite you to submit a revised version of the manuscript that addresses the points raised during the review process.

We look forward to receiving your revised manuscript.

Kind regards,

Obasanjo Afolabi Bolarinwa, Masters

Academic Editor

PLOS ONE

Journal Requirements:

Reviewers' comments:

Reviewer's Responses to Questions

**Comments to the Author**

1. If the authors have adequately addressed your comments raised in a previous round of review and you feel that this manuscript is now acceptable for publication, you may indicate that here to bypass the “Comments to the Author” section, enter your conflict of interest statement in the “Confidential to Editor” section, and submit your "Accept" recommendation.

Reviewer #2: All comments have been addressed

Reviewer #3: All comments have been addressed

2. Is the manuscript technically sound, and do the data support the conclusions?

Reviewer #2: Yes

Reviewer #3: Yes

3. Has the statistical analysis been performed appropriately and rigorously? 

Reviewer #2: Yes

Reviewer #3: (No Response)

4. Have the authors made all data underlying the findings in their manuscript fully available?

Reviewer #2: Yes

Reviewer #3: Yes

5. Is the manuscript presented in an intelligible fashion and written in standard English?

Reviewer #2: Yes

Reviewer #3: Yes

6. Review Comments to the Author

Reviewer #2: The observed comments has been included to enrich the quality of the manuscript. However, the authors has failed to state clearly the contribution this work has made to scholarly knowledge. Although, it has emphasized that lack of knowledge of the ovulatory cycle is a leading cause of unwanted pregnancy. Is this not a general knowledge? I will appreciate a clear cut contribution to scientific knowledge from this work, which is absolutely missing

Reviewer #3: (No Response)

7. PLOS authors have the option to publish the peer review history of their article (what does this mean?). If published, this will include your full peer review and any attached files.

Reviewer #2: **Yes: **chukwudeh Okechukwu Stephen

Reviewer #3: **Yes: **David Bamidele Olawade

---

## [Decision Letter · Decision Letter 2]

29 Jan 2024

PONE-D-23-11422R2Assessing the geographical variations in ovulatory cycle knowledge among women of reproductive age in Sierra Leone: Analysis of the 2019 Demographic and Health SurveyPLOS ONE

Dear Dr. Gbagbo,

Thank you for submitting your manuscript to PLOS ONE. After careful consideration, we feel that it has merit but does not fully meet PLOS ONE’s publication criteria as it currently stands. Therefore, we invite you to submit a revised version of the manuscript that addresses the points raised during the review process.

**ACADEMIC EDITOR: ****Please respond to all reviewers comments**==============================

We look forward to receiving your revised manuscript.

Kind regards,

Ahmed Mohamed Maged, MD

Academic Editor

PLOS ONE

Reviewers' comments:

Reviewer's Responses to Questions

**Comments to the Author**

1. If the authors have adequately addressed your comments raised in a previous round of review and you feel that this manuscript is now acceptable for publication, you may indicate that here to bypass the “Comments to the Author” section, enter your conflict of interest statement in the “Confidential to Editor” section, and submit your "Accept" recommendation.

Reviewer #2: All comments have been addressed

Reviewer #4: (No Response)

2. Is the manuscript technically sound, and do the data support the conclusions?

Reviewer #2: Yes

Reviewer #4: Partly

3. Has the statistical analysis been performed appropriately and rigorously? 

Reviewer #2: Yes

Reviewer #4: Yes

4. Have the authors made all data underlying the findings in their manuscript fully available?

Reviewer #2: Yes

Reviewer #4: Yes

5. Is the manuscript presented in an intelligible fashion and written in standard English?

Reviewer #2: Yes

Reviewer #4: No

6. Review Comments to the Author

Reviewer #2: This is a nice study that examined the link between ovulation and maternal mortality/NMR, and other pregnancy related complications.

It will be nice to add how many respondents were interviewed per household

Also add the policy implications of this study

Reviewer #4: Title: Assessing the geographical variations in ovulatory cycle knowledge among women of reproductive age in Sierra Leone: Analysis of the 2019 Demographic and Health Survey

General comments:

- Overall, the manuscripts lack an academic writing tone e.g. the manuscript is not attractive. The way the introduction of the manuscript was drafted is poor, the method should have been put in comparison to traditional methods or the advantage of the selected method over others, results should have clearly stated positive and negative associations separately, and conclusions should have more practical and actionable rather than showing directions. Thus, it would help if you redrafted the abstract.

Introduction

- I don’t think you can discuss this issue like you presented in the introduction section.

- In the first place you need to define what ovulatory cycle knowledge is

- For whom it is important

- I think you need a strong argument to show the topic is still valid in the presence of modern contraceptive methods. This is more of a traditional method and we do not recommend people follow it because of the costly mistakes, and differences in women's body physiology as the symptoms and signs you mentioned can only occur in some women. I won’t say it is not searchable but you need to play in the context and give enough ground to prove it important.

- Starting from the abstract to the introduction you need to answer the why questions, where is the gap, do we need this information why, and where will this information be applicable?

- Is this information important for policy decisions? Who is going to use this information why since modern contraception has better applicability?

- Thus, redraft the introduction section to fit your aim, and in this way, it will reveal what I can't see now from the existing context.

-

- Methods, results, and discussion

- Say something about DHS. Which type of DHS is the data coming from is that mini or main?

- How data customized for analysis should be discussed

- How missing data handled should be presented more clearly

- Results should be summarized in the same format and follow what was described in the method section

- Discussion should be formatted considering the contest requested above

7. PLOS authors have the option to publish the peer review history of their article (what does this mean?). If published, this will include your full peer review and any attached files.

Reviewer #2: **Yes: **Chukwudeh Okechukwu Stephen

Reviewer #4: **Yes: **Girma Gilano

---

## [Author Response · Author response to Decision Letter 2]

15 Feb 2024

Dear Editor,

REBUTTAL.

We have carefully looked at the comments from Reviewer #4 and have the following major concerns with his/her comments which questions the level competency, experience and knowledge of reviewer #4 in this area of study:

1. All the comments of reviewer #4 are a very subjective statements which do to add any value to our manuscript. It must be indicated that the authors are seasoned academics with sound academic writing skills. We think the reviewer #4 should have been more constructive in the choice of words. 

2. The reviewer #4 comments clearly show that he/she did not read the manuscript carefully. Most of the comments being made so far are not based on careful review of our manuscript. This particular comment about definition of ovulatory cycle knowledge has been well captured and explain in the introduction paragraph and with clear citations as follows: (‘Ovulation generally occurs at about the midpoint of the menstrual cycle. Therefore, for the average menstrual cycle of 28 days, ovulation will occur about 14 days after the onset of menstruation (i.e., when the period begins) [4]. It is important to note however that there can be variation in menstrual cycle length, which means that the occurrence of ovulation can vary from the typical 14-day mark [5]. In any case, menstruation can be used to estimate when one will ovulate [3]. Women who have a basic knowledge of the aforementioned physiological phenomena are better able to time their pregnancies, or avoid pregnancy if that is their goal [2,3]’).

3. Also think the reviewer #4 lacks in-depth knowledge about modern contraceptive methods particularly those that relate to natural family planning. In the light of current literature on contraception/family planning, the topic is still valid in the presence of modern contraceptive methods since not everybody is comfortable using ‘artificial’ contraceptive. 

We therefore disagreed with his/her comment since literature has shown that despite the failure rate, knowledge of ovulation cycle is very effective for pregnancy prevention for those who know how to use this method correctly. Besides not every contraceptive has 100% success rate.

Based on the above observations and in order not to unnecessarily further delay the publication of this paper, we strongly oppose the comments of reviewer #4 and highly recommend that our paper is discontinued from his/her review. 

We also wish to state that the previous three (3) reviewers who looked at our work under the supervision of the former editor who handled this manuscript were almost at a consensus to accept the paper for publications after satisfactory response to only the reviewer #2 comments. Example is the reviewer #2 who initially looked at our work and required for a minor clarity.

Thank you and we count on your usual corporation. 

Dr. Gbagbo

Corresponding Author

---

## [Decision Letter · Decision Letter 3]

26 Feb 2024

Assessing geographical variation in ovulatory cycle knowledge among women of reproductive age in Sierra Leone: Analysis of the 2019 Demographic and Health Survey

PONE-D-23-11422R3

Dear Dr. Gbagbo,

We’re pleased to inform you that your manuscript has been judged scientifically suitable for publication and will be formally accepted for publication once it meets all outstanding technical requirements.

Kind regards,

Ahmed Mohamed Maged, MD

Academic Editor

PLOS ONE

Additional Editor Comments (optional):

Reviewers' comments:

Reviewer's Responses to Questions

**Comments to the Author**

1. If the authors have adequately addressed your comments raised in a previous round of review and you feel that this manuscript is now acceptable for publication, you may indicate that here to bypass the “Comments to the Author” section, enter your conflict of interest statement in the “Confidential to Editor” section, and submit your "Accept" recommendation.

Reviewer #4: All comments have been addressed

2. Is the manuscript technically sound, and do the data support the conclusions?

Reviewer #4: Yes

3. Has the statistical analysis been performed appropriately and rigorously? 

Reviewer #4: Yes

4. Have the authors made all data underlying the findings in their manuscript fully available?

Reviewer #4: Yes

5. Is the manuscript presented in an intelligible fashion and written in standard English?

Reviewer #4: Yes

6. Review Comments to the Author

Reviewer #4: no further comment for this specific version. however, the way they responded to reviewer four is not something expected

7. PLOS authors have the option to publish the peer review history of their article (what does this mean?). If published, this will include your full peer review and any attached files.

Reviewer #4: No

---

## [Editor Report · Acceptance letter]

7 Mar 2024

PONE-D-23-11422R3 

PLOS ONE

Dear Dr. Gbagbo, 

I'm pleased to inform you that your manuscript has been deemed suitable for publication in PLOS ONE. Congratulations! Your manuscript is now being handed over to our production team.

Kind regards, 

on behalf of

Professor Ahmed Mohamed Maged 

Academic Editor

PLOS ONE